# Stem Cells for Cancer Therapy: Translating the Uncertainties and Possibilities of Stem Cell Properties into Opportunities for Effective Cancer Therapy

**DOI:** 10.3390/ijms24021012

**Published:** 2023-01-05

**Authors:** Ahmed Faris Aldoghachi, Zhi Xiong Chong, Swee Keong Yeap, Soon Keng Cheong, Wan Yong Ho, Alan Han Kiat Ong

**Affiliations:** 1M. Kandiah Faculty of Medicine and Health Sciences, Universiti Tunku Abdul Rahman, Kajang 43000, Malaysia; 2Division of Biomedical Sciences, School of Pharmacy, Faculty of Sciences and Engineering, University of Nottingham Malaysia, Semenyih 43500, Malaysia; 3China-ASEAN College of Marine Sciences, Xiamen University Malaysia, Sepang 43900, Malaysia; 4National Cancer Council (MAKNA), Kuala Lumpur 54000, Malaysia

**Keywords:** MSC, iPSC, iMSC, CSC, exosomes, miRNA, paracrine factors

## Abstract

Cancer recurrence and drug resistance following treatment, as well as metastatic forms of cancer, are trends that are commonly encountered in cancer management. Amidst the growing popularity of personalized medicine and targeted therapy as effective cancer treatment, studies involving the use of stem cells in cancer therapy are gaining ground as promising translational treatment options that are actively pursued by researchers due to their unique tumor-homing activities and anti-cancer properties. Therefore, this review will highlight cancer interactions with commonly studied stem cell types, namely, mesenchymal stroma/stem cells (MSC), induced pluripotent stem cells (iPSC), iPSC-derived MSC (iMSC), and cancer stem cells (CSC). A particular focus will be on the effects of paracrine signaling activities and exosomal miRNA interaction released by MSC and iMSCs within the tumor microenvironment (TME) along with their therapeutic potential as anti-cancer delivery agents. Similarly, the role of exosomal miRNA released by CSCs will be further discussed in the context of its role in cancer recurrence and metastatic spread, which leads to a better understanding of how such exosomal miRNA could be used as potential forms of non-cell-based cancer therapy.

## 1. Introduction

Over the past decade, registered clinical trials involving the use of MSC, IPSC, and CSC in cancer therapy have grown, and more recently, iMSC was added to the list. In the midst of the controversial reputation of MSC cells in the world of cancer studies, the therapeutic roles such as their tumor-homing capabilities, paracrine signaling activities, and delivery of exosomal microRNA (miRNA) cargo are well-known mechanisms employed by these cell types in communicating with cancer cells within the tumor microenvironment (TMC). In addition, with more reliable iPSC generated through non-viral and non-integrated reprogramming platforms, a breath of new options for more stable pluripotent cells and their derivatives are obtainable. One such candidate is MSC generated though differentiation of fully reprogrammed cells (iMSC) which may potentially offer another alternative cell type that compensates for the heterogeneity and scalability issues of MSCs while maintaining all the valuable therapeutic properties of MSCs. IPSCs have also been shown as feasible disease models for understanding cancer progression and their resistance to cancer drugs. Finally, by understanding the plasticity of CSC in tumor progression and metastasis, and how exosomal miRNA plays a pivotal role in the cancer stem cell niche, potential anti-cancer candidates could be developed, leading to non-cell-based cancer therapy. Therefore, this review focuses on the current perspective of how major stem cell types interact with cancer cells and their recent progress toward the development of cell-based and non-cell-based cancer therapy.

## 2. Mesenchymal Stroma/Stem Cells (MSC) and Cancer

Human MSCs are heterogenous populations of multipotent stem cells that can be obtained from different origins, and among the most commonly studied are the bone marrow, adipose tissue, umbilical cord, Wharton’s jelly, dental pulp, and peripheral blood mononuclear cells [1]. The international society for cell and gene therapy (ISCT) defined MSC as being able to adhere to plastic, positively express MSC markers (CD73, CD105, and CD90), negatively express hematopoietic markers (CD14, CD45, CD34, CD19, and HLA-DR), and have the ability to differentiate into adipogenic, chondrogenic, and osteogenic lineages [2]. ISCGT also suggested that the termed mesenchymal stromal cells be used instead of mesenchymal stem cells for any fibroblast-like plastic-adherent cells, regardless of the tissue from which they are isolated, while keeping the same acronym “MSCs” [3].

### 2.1. MSC Homing and Paracrine Interaction with Cancer

While studies have shown the various roles of MSCs in cellular functions, including the ability to differentiate into various cell types for damaged tissue repair and regeneration, current evidence has also pointed to the paracrine signaling mechanism [4,5] related to the release of several trophic factors including cytokines, chemokines, and extracellular matrix protein as well as exosomal miRNA into the surrounding cell environment. These paracrine signals were reported to be associated with immunomodulatory properties, angiogenesis, anti-apoptosis, anti-oxidation, anti-inflammation, and cell proliferation [4]. However, in the context of cell cancer interaction, paracrine signaling also promotes the migration of MSC to the tumor site [6] and has been known to either exert suppression or promotion of tumors within the tumor microenvironment (TME) [7]. Among the signaling molecules involved with MSC migration to the tumor site are CXCL12/CXCR4 [8], CCL2 in Breast cancer, and SDF-1 in colorectal, prostate, and breast cancers [9]. The recruitment of MSC to the cancer site can also be achieved via the interaction of MSC with certain cytokines responsible for angiogenesis (IL-8, TGF-β, and VEGF) that are secreted by cancer cells [10]. In addition, MMP-1, a component of the extracellular matrix, helps in stimulating MSC homing to the cancer site via PAR-1 cleavage and activation [11].

### 2.2. The Double Life of MSC in Cancer Interaction Studies

The dual role of MSC in cancer-related studies, as either promoting or suppressing cancer activities in cancer cells, is a much-accepted fact, with various reports linking MSC as a tumor enhancer via mediating angiogenesis, attenuating immune reactions, initiating epithelial to mesenchymal transition (EMT), and promoting metastatic processes whereas the tumoricidal effect of MSC on cancer cells appears to be more apparent through the induction of apoptosis and signaling pathways alterations [12].

#### 2.2.1. MSC as a Tumor Enhancer

Several studies have revealed that MSCs can support the tumor vasculature via direct differentiation of MSCs into myofibroblasts that are were then transformed into tumor-associated fibroblasts (TAFs) [13] or indirectly via the secretion of several growth factors [14]. MSCs are capable of secreting several angiogenic factors (FGF-2, PDGF, VEGF, TGF-β, IL-6, IL-8, and angiopoietin-1) that facilitate angiogenesis leading to tumor-promoting features [15]. Additionally, the immunosuppressive effect of MSC contributes to cancer cells’ escape from the immune system surveillance. Due to their direct action on immune cells, MSCs are capable of inhibiting apoptosis or inhibition of T cell proliferation which results in a decline of immunogenic activity [16]. Furthermore, MSC has also shown pro-metastatic effects that are mediated via paracrine factors, including TGF-β, CXCR4, and CCL5; chemokines secreted from MSC [10,16,17]. Martin et al. (2010) revealed that the co-culture of bone marrow-derived MSC with breast cancer resulted in the overexpression of EMT-specific genes and a decrease of mesenchymal to epithelial (MET) genes [18]. Similarly, in a co-culture-based interaction study between an adipose-derived MSC (ADSC) and MCF-7 breast cancer and in an in vivo nude mouse model, MCF-7 cells were shown to exert tumor tropism effects on ADSCs, reportedly regulated by chemokines, such as the macrophage inflammatory protein (MIP)-1δ and MIP-3α. This effect was mediated by epithelial–mesenchymal transition, which significantly induced tumor sphere formation in vitro and promoted tumorigenicity in vivo [19]. The interaction of MSC with cancer cells can also result in the differentiation of MSC into carcinoma-associated fibroblasts (CAFs) via TGF- β resulting in stabilizing the tumor tissue at the primary and metastatic site and thus promoting cancer stemness as well as chemoresistance via paracrine factor secretions [20]. MSC-conditioned medium has been shown to upregulate BCl-2, an anti-apoptotic protein, and suppress the expression of P53 and BAX (apoptosis proteins) that resulted in inhibiting apoptosis, which in turn promoted colorectal cancer progression through AMPK/mTOR-mediated NF-κB activation [21].

#### 2.2.2. MSC as a Tumor Suppressor

Mesbah et al. (2021) showed that the cultivation of colorectal cancer cells with MSC inhibited proliferation and induced apoptosis of colorectal cancer cells with increased expressions on P53, Caspase3, and P21. The tumoricidal or pro-neoplastic effect of MSC on cancer can also be attributed to its hypo- or hyper-activation of certain signaling pathways associated with cancer proliferation, survival, and progression [22]. Among the pathways involved are the ERK1/2, Wnt, PI3K/AKT, JAK/STAT, MYC, Hippo, and NF-kB pathways [23]. MSC-conditioned medium when cultured with head and neck squamous cell carcinoma cell lines resulted in enhancing cellular proliferation via the activation of ERK1/2 signaling in vitro [24]. Additionally, MSC administration into U251 glioma cells suppressed tumor growth and induced apoptosis via downregulating PI3K/AKT pathway [25]. Interestingly, a co-culture experiment on adipose-derived MSC (AD-MSC) with MCF7-luminal and MDA-MB-231-basal breast cancer cells demonstrated the possible influence of exosomal miRNA in promoting these cancer cells towards a more dormant-epithelial phenotype associated with lower metastasis potential but higher chemoresistance. The decrease in metabolic and cellular activities of these dormant cells was attributed to a possible form of evasion by the cancer cells from the effects of these drugs, which generally targeted rapidly proliferating cells, as well as the increased expression of drug resistance-related proteins triggered by MSC-exosomes [26].

#### 2.2.3. Exosomal miRNA in Crosstalk between MSC and Cancer Cells

Another exciting development at the forefront of MSC and cancer therapy is the interaction between exosomal miRNA and cancer cells. The dysregulation of exosomal miRNA was shown to promote tumor growth due to the activation of angiogenic signaling pathways such as VEGF [27], hedgehog signaling pathway [28], and STAT3 [29]. Other studies reported on the effects of exosomal miRNA that decreased tumor growth were associated with NF-κB p65 activation [30], downregulation of VEGF expression [31], inhibition of Galectin-3 [32], regulation of KDM4B/HOXC4/PD-L1 axis [33], down-regulation of mTOR and S6KB1 expression [34], and down-regulation of trefoil factor 3 (TFF3) [35]. Examples of cancer studies involving exosomal miRNA are summarized in Table 1, and a comprehensive list of miRNA shown to be associated with various cancer types was previously reviewed by Galland et al. [7] as well as Dalmizrak and Dalmizrak [36]. Despite the complexities of MSC interaction with cancers in the respective TME, numerous exosomal miRNA were found to be directly associated with major cancer pathways (Table 1) and were shown to be promising anti-cancer delivery agents with potential as non-cell-based cancer therapy (Table 2).

## 3. MSC and Its Derivatives as a Carrier for Anti-Cancer Agents

Due to the homing properties of MSCs towards the tumor site, recent studies have focused on utilizing MSCs as carriers for anti-cancer agents, including MSC expressing IL-18, TRAIL, oncolytic adenovirus (CRAd5/F11), and paclitaxel-encapsulated nanoparticles in the treatment of breast cancer cells [52], B-cell acute lymphocytic leukemia [53], colorectal cancer [54], and lung cancer [55] respectively. These interventions resulted in the inhibition of cancer proliferation and metastasis, induced apoptosis, and improved survival of patients. Such promising results opened a new approach to targeting cancer cells via expressing certain key therapeutic factors, which can diminish the divergent therapeutic potential of naive MSC in cancer treatment. Current reported clinical trials of anti-cancer therapies are in the form of tissue-derived MSC, engineered MSC (as a carrier of therapeutic oncolytic viruses or cytokines|) and MSC-derived exosomes [23].

Although a big proportion of trials registered on ClinicalTrials.Gov were designed to evaluate MSCs that are not genetically modified, limited data have since been published from such trials. One such published study described the role of allogenic bone marrow-derived MSC in localized prostate cancer patients to assess the safety and feasibility of MSC-homing capacity to localized prostate cancer regions (NCT01983709) and revealed that MSCs were as well tolerated with no dose-limiting toxicities observed. However, as MSCs were not detected to home primary tumors in any of the study subjects, the trial was terminated [56]. Nevertheless, considering the pro-tumor effects of MSCs in the TME and the controversial roles that MSCs play in the interactions with cancer cells, the administration of unmodified MSCs might not be an efficient way to treat cancers. Hence, clinical trials incorporating genetically engineered MSC in cancer have been carried out of which two published data revealed the delivery of autologous MSC expressing herpes simplex virus-thymidine kinase and oncolytic adenovirus administered to study subjects with gastrointestinal adenocarcinoma (EudraCTnumber: 2012-003741-1) [57,58] and relapsed pediatric solid tumors (NCT01844661) [59] respectively, were well tolerated, showed disease stabilization, and were reported to be safe for use by patients.

Furthermore, exosomes have been shown to be abundantly produced in MSCs [60], and exosomes from MSC are superior drug delivery methods when it comes to gene transfer capacity, biocompatibility, immunogenicity, and stability [61]. Therefore, another promising approach would be to use non-cell-based therapy, namely exosomes, as nanoparticles, for more efficient delivery of anti-cancer load such as anti-oncomirs. Notably, as an up-and-coming anti-cancer therapy method, so far, one promising MSC-derived exosome trial (NCT03608631) was reportedly designed to evaluate the safety and efficacy of MSC derived exosomes with KrasG12D siRNA (iExosomes) in pancreatic cancer [23]. Previous studies on exosomal miRNA with potential as anti-cancer agents are summarized in Table 2.

## 4. Challenges and Opportunities for MSC as a Cancer Therapeutic Agent

The dual roles of MSC to either suppress or promote tumor development could be attributed to the variations in the tumor model utilized among different studies, tissue source of MSC, epigenetic variability as well as the heterogeneity of the isolated MSCs., In addition, the timing and dosage of MSC administration, variations in cellular delivery methods and, culture conditions utilizing different supplements were commonly shown to influence the interaction between MSCs and cancer cells [62,63]. Furthermore, utilizing MSC from bench to bedside in cancer therapy is complex as several additional factors have to be considered, such as the patient’s cancer stage and pathological condition, and donor variations (genetics, gender, age, and health status) that may attribute to the induction of molecular variations in MSCs, rendering the therapeutic effect as inconclusive [64,65]. As such, it is evident that for MSC-based cancer therapy to remain applicable and relevant as a mainstream cancer therapy, a more homogenous form of MSCs with specific tumor-homing activity and more efficient drug delivery strategies to cancer cells are needed. [66,67] Although much is still unknown about the mechanisms by which exosomes and their miRNA interact with tumors, the advancement of gene engineering techniques provides opportunities to further develop these nano-size vesicles with improved targeting specificity for the delivery of anti-oncomirs [68]. For instance, exosomes have been shown to possess homing capabilities similar to that of MSC from which they were derived, and genetic modifications of membrane proteins on the exosomal surface could enhance its tumor targeting properties allowing the interaction of anti-oncomirs to be delivered in a more targeted manner [69]. These modifications include cell or tissue-specific peptides, tumor-specific receptors, and antibodies, among others [70]. Furthermore, the scalable propagation of MSCs for much more feasible exosome harvest and efficacy was shown to be possible with the use of three-dimensional extracellular matrix-based scaffolds commonly derived from natural sources or from synthetic biomaterials. As for large-scale expansion of MSCs, spinner flask culture with microcarriers, as well as hollow fibers bioreactors, have been reported to provide the capacity of cells needed for sufficient amounts of exosomes to be generated for clinical applications [71]. The engineering of exosomal loading with anti-cancer drugs may provide better-targeted effects on controlling tumor growth. One such study using engineered exosomes isolated from MB-MSC that were loaded with galectin-9 siRNA showed enhanced level of tumor targeting efficacy and tumor suppressive effects on a pancreatic cancer model [72].

## 5. Application of Induced Pluripotent Stem Cells (iPSC) in Cancer Studies

iPSCs were first discovered in 2007 by [73], whereby through the reprogramming of adult somatic cells via the introduction of four specific pluripotent-associated genes, including Oct3/4, Sox2, KIF4, and c-Myc. The process of reprogramming occurs via the transduction of pluripotent-associated genes into cells using integrative or non-integrative systems along with either viral or non-viral delivery vectors. The generated iPSCs resembled the embryonic stem cell (ESC)-like morphology, express ESC markers (SSEA-1, Nanog), and form teratomas upon injection in immunocompromised mice. iPSCs possess a high self-renewal capacity and proliferation; differentiation to mesoderm, endoderm, and ectoderm, and bypassed ethical concerns that arise from the usage of human embryonic stem cells (hESCs) [74].

### 5.1. iPSC Derived MSC (iMSC) and Cancer Therapy

Common issues that affect the clinical translation of using MSC in cancer therapy are its heterogeneity and difficulty of scalability [75]. Hence, a newer approach focusing on the bulk generation of induced MSCs (iMSCs) via differentiation of induced pluripotent stem cells (iPSCs) may provide a more practical solution for clinical applications. iMSCs can be generated in abundance via iPSCs differentiation and these cell type incorporates the advantages of both MSCs and iPSCs with no or low immunogenicity [76], capable of being cultured under feeder free system [77] and maintenance of their MSC characteristics without any chromosomal abnormalities even at prolonged passage [78]. Current evidence also supports iMSC as being more superior than primary MSC of various tissue sources in terms of the molecular signature, proliferation capacities, tissue repair, and differentiation applications [79]. Furthermore, with more improved reprogramming methods namely non-viral integration and omitting transcription factors of a tumorigenic gene such as c-myc, the risk of genomic instability and tumor transformation potential are minimized, making scalable sources of iMSC available as suitable candidates for personalized cancer therapy [75].

Several in vitro and in vivo studies have been carried out to test the impact of iMSC on multiple cancers including breast [80] and lung cancers [81]. In a study to compare interaction of BM-MSC and iMSC on with BC, it was found that the efficiency of iMSC homing to cancer was similar to that of BM-MSC. However, iMSCs were less prone to promoting EMT, proliferation, and invasion of BC cells. These results were attributed to the low expression of IL-1, TGF-B, and TSG6 in iMSC cells compared to that of BM-MSC [80].

Further studies with genetically engineered iMSC revealed promising therapeutic effects on different cancers [82,83,84]. Liu et al. revealed that iMSC-IL24 induced apoptosis and inhibited the growth of B16-F10 melanoma cells with higher efficiency compared to that of iMSC alone either through an in vitro or an in vivo platform [82]. Additionally, the incorporation of TRAIL in iMSC (iMSC-TRAIL) via site-specific integration into ribosomal DNA resulted in inducing apoptosis in melanoma, liver, breast, and lung cancers in vitro and inhibiting tumor growth through the activation of apoptotic pathways in vivo [83]. Furthermore, Portier et al. generated iMSCs from the PBMC of two individuals carrying the BRCA1 [(iMSC-BRCA1(-)] mutations and when compared to the iMSC generated from iMSC of normal controls, iMSC-BRCA1(-) exhibited pro-angiogenic signature via overexpressing angiogenic factors such as VEGF, PDGF, ANGPT, and HIF-1α. In addition, iMSC-BRCA1(-) had an increased capacity over normal controls for generating tube-like structures and vessels in vitro and in vivo [84].

Notably, extracellular vesicle (EV) mimics made from iPS cell-derived mesenchymal stem cells were shown to improve the treatment of metastatic prostate cancer [85] and generate more targeted therapeutic possibilities as anti-cancer delivery agents [86]. These promising findings demonstrated a possible therapeutic aspect of cancer therapy that has since proven worthy of clinical trials. Currently, there has been one Phase I data published on the potential utility of iMSC on steroid-resistant graft versus host disease, which revealed a promising outcome, whereby iMSC was safe and well tolerated with no observable adverse reactions in patients [87]. Another Phase I/II trial is being conducted to determine the potential role of iMSC-CYP-001 on patients with respiratory failure involving COVID-19 to assess respiratory dysfunction among groups administered with iMSC-CYP-001 versus standard of care treatment group (NCT04537351).

### 5.2. iPSCs as a Cancer Model

iPSCs can be generated from malignant cancer cells as a live human cell model to evaluate the stages of oncogenesis, which starts with cellular transformation followed by the hierarchical organization of established malignancies. Also, reprogramming cancer cells could aid in determining the interaction among the genetic and epigenetic drivers of carcinogenesis. In a colorectal cancer model, iPSCs-derived colorectal cancer organoids were developed to test drug sensitivity, revealed that geneticin was able to suppress WNT signaling activation, rescue adenomatous polyposis coli (APC) protein expression levels, and suppressed the colonic epithelial proliferation in familial adenomatous polyposis colonic organoid model [88]. In another study, iPSCs derived from Li-Fraumeni syndrome (LFS) patients were differentiated into MSC and then into osteoblasts (cells of origin for osteosarcoma) which showed oncogenic signatures of osteosarcoma that can aid in understanding the pathogenesis of osteosarcoma [89]. In an interesting study using iPSC-derived cardiomyocytes to determine the potential role of trastuzumab in inducing cardiac dysfunction on breast cancer patients, it was found that trastuzumab impaired the contractile properties of iPSC-derived cardiomyocytes without causing cardiomyocyte death. Furthermore, RNA-Seq analysis revealed that trastuzumab altered the cardiac energy metabolism pathway, resulting in cardiac toxicity. It was concluded that patient-derived iPSC-cardiomyocytes with severe cardiac dysfunction were more vulnerable to trastuzumab compared to iPSC-cardiomyocytes derived from patients with no cardiac dysfunction following therapy [90].

### 5.3. Therapeutic Potentials of iPSC

Alternatively, iPSCs may serve as another avenue for adoptive immune therapy via its deviation into natural killer cells (iPSCs-NK). These iPSCs-NK cells could be engineered via knock-in/out of genes resulting in the generation of allogenic iPSCs-NK cells capable of suppressing MHC1 leading to avoidance of host T cell recognition. These engineered cells also over-expressed HLA-E, resulting in the inactivation of host NK cells [91]. The reprogramming of NK cells to iPSCs to express chimeric antigen receptor-NK cells (NK-CAR-iPSC-NK cells) has been shown to mediate a strong response in inhibiting tumor growth as well as enhanced in vitro and in vivo cell survival in an ovarian cancer study model when compared to that of peripheral blood-NK cells [92].

## 6. Cancer Stem Cells and Their Exosomal miRNA

Cancer stem cell (CSC) is a group of highly dysregulated cancer cells with abnormal karyotyping, unlimited capacity to self-proliferate, highly aggressive, and possesses abilities to resist different types of anti-cancer therapies [93,94,95,96,97]. Depending on the cancer type, different CSC surface markers have been reported to associate with specific CSC [93] and examples of such CSC surface markers include CD44 (breast CSC) [98] and CD133 (glioma CSC) [99]. In recent years, circulating miRNAs, especially exosomal or extracellular vesicle (EV)-encapsulated miRNAs have been reported to play essential roles in modulating the cancer cell stemness of many cancer types [97,100,101,102]. Besides, the miRNAs identified in the exosomes or EV isolated from the tumor microenvironment (TME) or systemic circulation were also shown to have important roles in predicting the prognosis of cancer patients as some of these miRNAs were found to be involved in regulating cancer cell stemness and treatment responses in specific cancer types (Appendix A) [94,103,104,105].

### 6.1. Roles of Circulating CSC-Regulating miRNAs in Modulating Cancer Cell Stemness

The TME in breast cancer consists of cancer cells, immune cells, stromal cells, and various local factors originating from either the local breast cancer or distant tissues [106]. In breast cancer, miR-155 was found to be upregulated in the exosomes isolated from the breast CSC and chemoresistant cells [98], and thus, the measurement of exosomal miR-155 level isolated from the TME could help to predict treatment response among breast cancer patients. The overexpression of miR-155 was demonstrated to enhance the expression of genes that regulate epithelial-to-mesenchymal transition (EMT), such as SNAIL and SLUG [98]. The increased expression of SNAIL and SLUG have been reported to promote cellular proliferation, EMT, and cellular movement in the TME, and the net results include increased cancer metastases [107]. In another in vitro study [96], the administration of chemotherapy was shown to induce the release of EV-encapsulated miRNAs containing miR-9-5p, miR-195-5p, and miR-203a-3p and these miRNAs were demonstrated to target ONECUT2 (Figure 1). Suppression of ONECUT2 increased the expression of CSC-regulating targets such as OCT4 and SOX2 [96]. Eventually, the chemotherapy-induced release of CSC-promoting miRNAs would lead to enhanced stemness and chemoresistance in the breast cancer cells [96]. SFRP1 is an antagonist of the oncogenic WNT signaling pathway [108], and miR-93-3p and miR-105 were reported to suppress SFRP1 expression to promote breast cancer cell stemness, chemoresistance, and metastases [109]. The elevation of circulating miR-93-3p and miR-105 levels was proven to correlate well with poor prognosis among triple-negative breast cancer (TNBC) patients [109]. Another circulating miRNA, miR-130a-3p, was reported to be under-expressed in the exosomes isolated from breast cancer patients with advanced staging, and this miRNA was found to be able to suppress RAB5B to reduce the breast cancer cell stemness [110]. Even though the study did not report that the elevation of exosomal miR-130a-3p would increase lymphocyte infiltration or stromal cell proliferation in breast cancer patients, but the increased expression of this circulating miRNA was found to correlate tightly to distant lymph node metastases [110].

In colorectal cancer, tumor-infiltrating cells, extracellular matrix (ECM), blood vessels, and other matrix-associated substances make up the TME [113]. As a key cellular activity that promotes cancer cell stemness, the EMT process can also be manipulated by circulating CSC-regulating miRNAs in colorectal cancer [111]. An in vitro study has recently reported that delivery of exosomal miR-375-3p into colorectal cancer cells would suppress the EMT process by suppressing the β-catenin, SNAIL, and Vimentin expressions [111]. The increased = in the EMT activities speeds up the breaking down of the cellular basement membrane and ECM, and thus, promoting cancer cell metastases [107]. On the contrary, two other in vitro studies have shown that delivery of exosomal miR-30a, miR-222, and miR-146a-5p could promote colorectal cancer cell stemness by targeting two tumor-suppressor targets, namely, MIA3 and NUMB, respectively [114,115]. miR-30a and miR-222 were demonstrated to target MIA3 [115], while miR-146a-5p was reported to be suppressing NUMB expression in colorectal cancer cells [114]. FBXW7 is an antagonist to the β-catenin of the WNT signaling [116], and exosomal miR-19b has been reported to target and suppress FBXW7 expression in colorectal cancer by enhancing its stemness and promoting radioresistance (Appendix A) [97]. The increased WNT signaling activity has been reported to link to enhanced cellular proliferation and increased anti-tumor activity in the TME [117]; this explains how the enhanced WNT signaling activity leads to increased radioresistance in the colorectal cancer cells. Similarly, FBXW7 could also be downregulated by exosomal miR-500a-3p in gastric cancer and is linked to enhanced disease staging, chemoresistance, and poor survival among gastric cancer patients [94]. Other than affecting CSC features of specific cancer types, the exosomal miRNA profiles could also be employed as biomarkers that predict the prognosis of gastric cancer patients, and a combined clinical and in vitro study [118] has identified six exosomal miRNAs that are related to gastric CSC and these include miR-424-5p, miR-590-3p, miR-628-5p, miR-675-3p, miR-1246, and miR-1290. Therefore, the measurement of the

High-grade glioma is a highly aggressive brain tumor [112] and its TME contains glioma cells, immune cells such as macrophages, and other neuronal supporting cells that include astrocytes [119]. An in vitro study [112] that involved pediatric glioma stem cells found that at least 35 types of miRNAs were over-expressed in the exosomes isolated from the primary cell culture and these miRNAs could potentially act as glioma stem cell biomarkers. On the other side, EV-miR-30b-3p was reported to suppress RHOB expression in glioblastoma cells, and this promoted stemness and resistance to temozolomide (TMZ) in glioblastoma [120]. The IGF1/AKT pathway is an oncogenic signaling pathway that is involved in promoting stemness in certain cancer types [121]. miR-603 has been previously identified as a miRNA that could target both IGF1 and IGFR1 to suppress the IGF1/AKT pathway and glioblastoma stemness [104]. However, the introduction of radiotherapy was found to promote the cellular export of miR-603 in the form of EV from cancer cells and this led to the de-repression of both IGF1 and IGFR1 [104]. Eventually, this increased tumorigenesis and stemness of the glioblastoma cells [104]. On the other hand, EV-associated miR-504 was reported to decrease glioma cell stemness and reduce the tumor-promoting activities of the microglia and macrophages in the TME and this indicates that therapeutic delivery of miR-504 could help to slow glioma progression [102]. In another Australian study [95], eight EV-miRNAs that include miR-16-5p, miR-23a-3p, miR-144-3p, miR-155-5p, miR-320e, miR-363-3p, miR-495-3p, and miR-520f-3p were shown to target PTEN to enhance the signaling activity of the AKT pathway (Figure 1). Therefore, these eight miRNAs have the potential to promote tumorigenesis and radioresistance in glioblastoma [95]. Besides, PTEN could also be repressed by miR-19b-3p in clear cell renal cell carcinoma as the introduction of exosomal miR-19b-3p was found to promote EMT and stemness in renal cell carcinoma [122]. Like many other cancer types, the TME in renal cell carcinoma also contains renal cancer cells, matrix-associated substances, and other supporting cells, and the increase in the EMT activity leads to increased cancer cell invasion and metastases [123].

The TME in liver cancer typically consists of cancerous cells, stromal cells, vasculature and immune cells that form a microenvironment that promotes cancer cell proliferation, migration, and treatment resistance [124]. In liver cancer, miR-1246 was reported to target AXIN2 and GSK-3β [100], in which both targets are suppressors for the canonical WNT signaling pathway [125]. The suppression of WNT signaling activity has been shown to promote anti-tumor activities in the TME and treatment sensitivities in most cancer types, such as colorectal cancer [117]. A high plasma level of miR-1246 was detected in hepatocellular carcinoma patients, and this is correlated tightly to low survival and chemoresistance among the patients [100]. CD90 is a hepatocellular carcinoma CSC marker [126] and delivery of exosomal miR-125a and miR-125b have been reported to suppress CD90 expression in hepatocellular carcinoma cells to decrease their stemness [99]. mTOR is a downstream target of the AKT signaling pathway and an in vitro study has shown that the delivery of exosomal miR-210 could activate the mTOR signaling pathway in pancreas cancer to promote cancer cell proliferation, stemness, and gemcitabine resistance [105]. However, as pancreas cancer TME contains desmoplastic stroma and other immune cells that promote metastases and treatment resistance [127], the in vitro study [105] did not further report on whether exosomal miR-210 would play a vital role in modulating the immune activity in the pancreas cancer TME. Targeting the immunosuppressive pathway in pancreas cancer has been highlighted as a potential approach to eradicate treatment-resistant pancreas cancer [127] and future studies could probably focus on investigating whether miR-210 could modulate the immune cells’ proliferation and activity in the pancreas cancer TME. On the other hand, the introduction of exosomal miR-139-5p was reported to be overexpressed in the exosomes isolated from the prostate CSC [103]. MiR-139-5p was said to contribute to the formation of pre-metastatic niche in prostate cancer TME, and the miRNA could increase the expressions of several metalloproteinases that are important in modulating cancer cell stemness and metastases such as MMP2, MMP9, and MMP13 [103]. In oral squamous cell carcinoma (OSCC), miR-21 and miR-32 were demonstrated to link to decreased OSCC stemness as low levels of these two miRNAs were detected in the exosomes isolated from the OSCC stem cells population [128]. However, the downstream targets that could be repressed by both miR-21 and miR-32 were not further identified and reported in the mentioned study [128]. In addition, it was also unclear whether both miR-21 and miR-32 would alter the OSCC TME by affecting the stromal and immune cells’ composition and activities, and this area is something worthy of further exploration in future studies.

Like other solid tumors, TME does exist in hematological malignancies and TME in hematological malignancies typically contain various hematological cells, immune cells, and local factors that form a proliferative niche with enhanced pro-tumoral activities [129]. Several CSC-regulating miRNAs have been reported to play vital parts in modulating cancer cell stemness and treatment resistance in several liquid cancer [130,131,132]. MiR-9 was found to suppress HES1 in acute myeloid leukemia (AML) to promote leukemia cell proliferation, and HES1 level was significantly downregulated in leukemic stem cells [131]. RAB27B is a protein molecule that is involved in exosome secretion and miR-34c-5p was reported to be able to suppress RAB27B expression to increase its intracellular level [132]. As a result, a high level of miR-34c-5p would induce senescence and eradication of leukemic stem cells in AML, and this was found to promote the survival of AML patients [132]. In myelodysplastic syndrome and leukemia, miR-22 was shown to suppress TET2 expression, and this protein is essential to regulate the renewal of hematopoietic stem cells by catalyzing the formation of hydroxymethylcytosine (5-hmC) from 5-methylcytosine (5-mC) [133]. Therefore, the downregulation of TET2 by miR-22 would promote the proliferation of leukemic cells and lead to low survival among the patients [133]. In multiple myeloma, increased expression of miR-1305 was detected in the exosomes isolated from the multiple myeloma cultures, while a low intracellular level of miR-1305 was noted in the multiple myeloma cells [130]. As miR-1305 has been described to be playing a tumor-suppressing role by suppressing the expressions of IGF1, FGF2, and MDM2, the low intracellular level of miR-1305 would the poor prognosis among multiple myeloma patients and it was hypothesized that the exosomal release of miR-1305 was linked to the low survival among the multiple myeloma patients [130].

In summary, circulating CSC-regulating miRNAs could serve as double-edged swords in either promoting or suppressing cancer development [95,96,111,112]. The entry of the circulating CSC-regulating miRNAs into the cancer cells could potentially activate or inhibit various oncogenic signaling pathways, such as the PI3K/AKT/mTOR [95,104,105] and WNT signaling pathway [94,100], and this would eventually affect cancer cell proliferation, invasion, and metastasis [95,96,111,112]. Besides, some of the CSC-regulating miRNAs could also affect the TME by affecting the EMT processes [122], the breakdown of matrix-associated substances, or affect the cellular activities of immune or other stromal cells in the TME [102]. As a result, this may create a pro-tumoral and pro-metastatic niche that enhance cancer cell growth and metastasis [103]. Therefore, by identifying key circulating CSC-regulating miRNAs that play essential roles in modulating cancer cell stemness or TME, these miRNAs have golden values to be employed as prognostic cancer biomarkers [95,96,111,112].

### 6.2. Identification of Circulating CSC-Regulating miRNAs in Clinical Trials

To date, at least six clinical trials (Table 3) have aimed to identify circulating CSC-regulating miRNAs in different cancer types have been executed (https://clinicaltrials.gov/) (accessed on 30 November 2022). Among these studies, two trials (NCT01577511 and NCT02052908) have focused on colorectal cancer, while one trial has focused on AML (NCT01298414), breast cancer (NCT01231386), glioblastoma (NCT05328089), and melanoma (NCT01216787), respectively. Four out of the six studies have been completed, while one study (NCT05328089) is still recruiting participants, whereas one trial has been terminated because of unclear reasons. Some of these studies are aimed at identifying specific circulating miRNA expression profiles in the cancer stem cells (e.g., NCT01298414), and some other studies (e.g., NCT01231386 and NCT01577511) are working to delineate specific profile changes in the circulating miRNAs after an interventional therapy was administered.

### 6.3. Challengers and Future Direction

Circulating exosomal or EV-encapsulated miRNAs have great potential in mediating the CSC phenotypes in different types of cancers [94,97,132]. Besides, the release of certain exosomal or EV-encapsulated miRNAs into the TME or systemic circulation could also serve as important biomarkers that help to identify specific CSC and this have prognostic value in predicting treatment response and survival among cancer patients [95,112,118]. However, before these circulating CSC-regulating miRNAs could be employed as prognostic biomarkers in clinical settings, a few challenges have to be overcome. Firstly, most of the currently reported studies (Appendix A) are focusing on in vitro studies, and thus more large-scale clinical studies that involve the isolation of exosomal or EV-miRNAs from cancer patients should be conducted to further establish the prognostic roles of specific circulating miRNAs in mediating the stem cell features in certain cancer types. Secondly, the isolation method of exosomal or EV-miRNAs from human cancer patients needs to be less invasive and scalable, while maintaining the production of highly purified exosomal miRNAs [134]. With the advancement of nanoscale technology, the isolation, detection, and profiling of exosomes can be made more feasible to cater to their potential use as noncell-based cancer therapy [135].

## 7. Conclusions

There is no doubt that the knowledge gained from studies involving MSCs, iMSC, iPSCs, and CSC with the tumor microenvironment has enhanced our understanding of their unique stem cell properties, often showing plasticity in response to their surrounding environment. Despite the risk of propagating the side effects of cancer progression or tumor growth, these stem cells have also shown to be a promising anti-cancer agent especially through the natural paracrine signaling and exosomal miRNA interaction with cancer cells. However, there still exist many uncertainties on how these stem cells and their secretome cause a direct influence on inhibiting tumor growth or cancer progression to a metastatic state. Nevertheless, numerous findings of in vitro cell studies and clinical trials have provided a way forward with the prospect of using targeted approaches such as exosomal miRNA as an alternative to cell-based cancer therapy to generate more specific positive outcomes as well as bioengineering methods for a more safer and specific delivery of cell- and non-cell-based therapy along with overcoming scalability and heterogeneity of stem cells. These are certainly elements of promising avenues that garner a strong case for incorporating and leveraging on the available knowledge and technological advancement in the area of stem cell research and their feasible application in paving a sustainable approach towards generating good manufacturing practices (GMP) grade cancer therapies capable of delivering the appropriate anti-cancer drug in a more targeted manner with precision effect on cancer treatment.

## Figures and Tables

**Figure 1 ijms-24-01012-f001:**
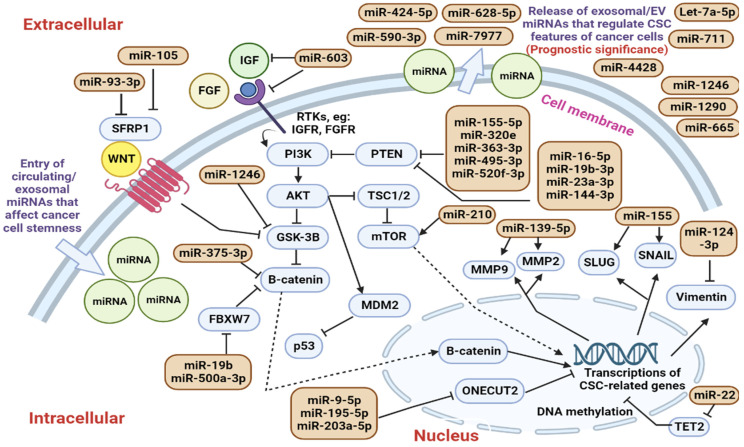
Interactions between different CSC-regulating miRNAs and their targets in modulating cancer cell stemness. Circulating, exosomal, or extracellular vesicle-encapsulated miRNAs could enter the cancer cells to regulate different signaling pathways that are involved in modulating cancer cell stemness [95,96,111,112]. Among the commonly reported pathways that are regulated by the circulating CSC-regulating miRNAs include the canonical WNT [94,100] and PI3K/AKT/mTOR [95,104,105] signaling pathways. The release of exosomal or EV-encapsulated miRNAs into the tumor microenvironment (TME) could serve as important biomarkers for detecting specific CSC, and this could have prognostic significance in predicting patient survival and treatment response to certain anti-cancer therapy.

**Table 1 ijms-24-01012-t001:** Studies on Exosomal miRNA involved in either tumor promotion or suppression.

Exosomal miRNA	MSC	Cancer Type	Signaling Pathway	Function	Ref.
mrR-21 miR-34a	Bone marrow	Breast cancer	Activation of (ERK1/2) pathway	Promote tumor growth	[27]
miR-221	Bone marrow	Osteosarcoma (MG63) and gastric cancer (SGC7901) cells	Activation of hedgehog signaling pathway.	Promote tumor growth	[28]
miR-193a-3p miR-210-3p miR-5100	Bone marrow grown under hypoxic condition	Lung cancer cells and an in vivo mouse syngeneic tumor model	STAT3-induced EMT	Promote cancer cell invasion and EMT.	[29]
miR-221	Bone marrow	Gastric cancer BGC-823 and SGC-7901 cells	ND	Proliferation, migration, invasion, and adhesion to the matrix of GC BGC-823 and SGC-7901 cells were significantly enhanced	[37]
miR-100-5p miR-9-5p let-7d-5p	Bone marrow	Glioblastoma	Activation of MSCs into (CAFs)-like cells	Promote tumor growth via a decrease in anti-tumoral miR-100-5p, miR-9-5p, and let-7d-5p	[38]
miR-17-5p miR-615-5p	Human adipose MSCs	Hepatocellular carcinoma cell line (Huh-7 cells)	Generation of cancer-associated phenotype of some CAF-like characteristics	Promote tumor growth via upregulation of miR-17-5p and 615-5p	[39]
miR-16	Bone marrow	Mouse breast cancer cell line (4T1)	Down-regulation of expressed VEGF in tumor cells	Suppress tumor growth via inhibition of angiogenesis	[31]
miRNA-1231	Bone marrow	Pancreatic cancer	ND	Suppress tumor growth	[40]
miRNA-16-5p	Adipose-derived mesenchymal stem cells	Breast cancer	ND	Suppress tumor growth	[41]
miRNA-128-3p	Human umbilical cord mesenchymal stem cell-	Pancreatic ductal cell carcinoma	Inhibiting galectin-3	Suppress pancreatic ductal cell carcinoma	[32]
miR-15a	Adipose-derived mesenchymal stem cells	Colorectal cancer	Restriction of immune evasion of CRC via the KDM4B/HOXC4/PD-L1 axis	Suppress tumor growth	[33]
miR-199a-3p	T-MScs	HepG2 cells.	Potentially targeting CD151, integrin α3 and 6	Inhibit tumor growth and HepG2 cell migration	[42]
miR-375	Enriched in bone marrow mesenchymal stem cells (BMSC)	Prostate cancer cell	Down-regulating trefoil factor 3 (TFF3)	Inhibit migration and invasion	[35]

Abbreviation: ERK1/2: extracellular signal-regulated kinase1/2; STAT3: signal transducer and activator of transcription 3; EMT: epithelial to mesenchymal transition; CAFs: cancer-associated fibroblasts; VEGF: vascular endothelial growth factor; KDM4B/HOXC4/PD-L1: lysine demethylase 4B/homeobox C4/programmed death-ligand 1; TFF3: trefoil factor 3; ND: not defined.

**Table 2 ijms-24-01012-t002:** Exosomal miRNA with potential as anti-cancer delivery agents.

Exosomal miRNA	MSC	Cancer Type	Delivery Method	Function/Target	Ref.
miR 222/223	ND	Immunodeficient mouse model of dormant breast cancer	MSC transfected with antagomiR 222/223	ND	[43]
microRNA-584	Human MSC (Origin ND)	U87 human glioma cells	Exosomes derived from microRNA-584 transfected mesenchymal stem cells	Suppression of the expression of CYP2J2; reduced the levels of phosphorylated AKT and MAPK	[44]
miRNA-221	Human cord blood mesenchymal stromal	Colorectal carcinoma	Cell-derived exosomes were used in the delivery of anti-miRNA oligonucleotides	Anti-tumor efficacy	[45]
miR-381-3p Mimic	Adipose-derived mesenchymal stem cells	MDA-MB-231 cells	miR-381 loaded ADMSC-exosomes	Downregulation of expressed related genes and proteins; inhibited proliferation, migration, and invasion capacity	[46]
miR-34a	Dental pulp MSCs (DPSCs)	Breast carcinoma cells.	miR-34a loaded modified dental pulp MSCs (DPSCs) exosomes	Repression of tumor proliferation	[47]
miR-30c-5p	Human umbilical cord mesenchymal stem cells	Papillary thyroid carcinoma (PTC)	miR-30c-5p containing extracellular vesicles	Tumor-suppressive miRNA targeted PELI1 to inhibit PTC cell proliferation and migration via activating PI3K/AKT pathway	[48]
Let-7f miRNA	Bone marrow-derived human mesenchymal stem cells	4T1 breast tumor model	Let-7f miRNA containing extracellular vesicles	Regulates SDF-1α- and hypoxia-promoted migration of mesenchymal stem cells	[49]
MiR-199a-	Adipose tissue-derived mesenchymal stem	Hepatocellular carcinoma	miR-199a-modified exosomes	Improve chemosensitivity through mTOR	[50]
miR-124	BM-MSC		miR-124 derived exosomes	Anti-tumor effects on cell proliferation, epithelial–mesenchymal transition, and chemotherapy sensitivity	[51]

Abbreviation: CYP2J2: cytochrome P450 2J2; AKT: serine/threonine kinase (protein kinase B); MAPK: mitogen-activated protein kinase; PELI1: pellino homolog 1; PI3K: phosphatidylinositol-3-kinase; SDF-1α: human stromal-derived factor 1; ND: not defined.

**Table 3 ijms-24-01012-t003:** Clinical trials that are aimed to identify circulating CSC-regulating miRNAs in different cancer types (n = 6).

Trial ID No.	Trial Title	Cancer Type	Study Location	Study Type (Sample Size, n)	Study Duration (Status)
NCT01298414	Pediatric Myeloid Leukemia-Specific miRNA Expression Profiles Induced by the Leukemic Stem Cell Niche	Acute myeloid leukemia	Feinstein Institute for Medical Research, United States of America	Observational, retrospective (20)	February 2011–May 2016 (Completed)
NCT01231386	MIRNA Profiling of Breast Cancer in Patients Undergoing Neoadjuvant or Adjuvant Treatment for Locally Advanced and Inflammatory Breast Cancer	Breast cancer	City of Hope Medical Center, United States of America	Observational (199)	November 2009–May 2019 (Completed)
NCT01577511	Invasiveness and Chemoresistance of Cancer Stem Cells in Colon Cancer: Molecular Characterization and Implications for Therapeutic Strategies	Colorectal cancer	Nîmes University Hospital, France	Observational, prospective (60)	June 2012–October 2017 (Completed)
NCT02052908	A Phase Ib Biomarker Trial of Naproxen in Patients at Risk for DNA Mismatch Repair Deficient Colorectal Cancer	Colorectal cancer	Brigham and Women’s Hospital; University of Michigan Comprehensive Cancer Center; M D Anderson Cancer Center; Huntsman Cancer Institute/University of Utah, United States of America	Interventional, randomized, double-blinded (81)	January 2014–January 2021 (Completed)
NCT05328089	Vacuolar ATPase and Drug Resistance of High-Grade Gliomas: a Study to Investigate Possible Therapeutic Roles for Proton Pump Inhibitors	Glioblastoma multiforme	University of Milano Bicocca, Italy	Observational, prospective (20)	January 2020–present (Recruiting)
NCT01216787	A Pilot Trial to Evaluate the Molecular Effects of RO4929097 as Neoadjuvant Therapy for Resectable Stage IIIB, IIIC, or IV Melanoma	Melanoma	Montefiore Medical Center; New York University Langone Medical Center, United States of America	Interventional, single group assignment (0)	September 2010–November 2011 (Terminated)

## Data Availability

No new data were created or analyzed in this study. Data sharing is not applicable to this study.

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
