# Peer review of "Stem Cells for Cancer Therapy: Translating the Uncertainties and Possibilities of Stem Cell Properties into Opportunities for Effective Cancer Therapy"

_ijms, 2023, doi:10.3390/ijms24021012_

Round 1
Reviewer 1 Report
The authors summarized the recent work on cancer interactions with commonly studied stem cell and their potential breakthroughs. The review is therefore timely but there are still some confusing points. I have a few comments that help to improve the clarity of the work. This review needs major revision before publication.
1. The manuscript needs to be checked again for grammatical and word spelling errors, and for chart formatting alignment issues.
2. In the line 129-132 of page 3, the aothors mention that “Interestingly, a coculture experiment on Adipose-derived MSC with MCF7-luminal and MDA-MB-231-basal breast cancer cells demonstrated the change towards a more dormant-epithelial phenotype associated with lower metastasis potential but higher chemoresistance [26].” I would like to ask if the authors could explain in more detail why the shift to a more dormant-epithelial phenotype would cause the cells to a higher chemoresistance.
3. In the chapter 2.2.3 of page 3-4, the end of the paragraph mentions that “these studies fell short of explaining ....”. I would like to know if the first half of the paragraph is your explanation of this issue? and if so, please adjust the order of the paragraph reasonably.
4. In the line 213-214 of page 8, there mentions “the advancement of gene engineering techniques provides opportunities to further develop”. Suggest that the authors consider where and what opportunities genetic engineering would open up and what possibilities it offers for subsequent specific applications.
5. In the line 307-320 of page 10 and Table S1, The authors summarize “Roles of circulating CSC-regulating miRNAs in modulating cancer cell stemness and its linkage to disease progression and prognosis” in detail. The work is very detailed and comprehensive, but it may be worth considering a brief description in the body content to summarize some warning points for the readers.
6. At the conclusion, the authors have summarized the current progress of stem cell for cancer therapy research and some difficulties that need to be overcome, and it would be nice if the authors could offer some more constructive comments after your own consideration.
7. The authors may consider adding relevant references to add to the richness of this work on exosomal anticancer service studies. For example: “Progress in the research of nanomaterial-based exosome bioanalysis and exosome-based nanomaterials tumor therapy. DOI: 10.1016/j.biomaterials.2021.120873”.
Author Response
Reviewer 1
The authors summarized the recent work on cancer interactions with commonly studied stem cell and their potential breakthroughs. The review is therefore timely but there are still some confusing points. I have a few comments that help to improve the clarity of the work. This review needs major revision before publication.
- The manuscript needs to be checked again for grammatical and word spelling errors, and for chart formatting alignment issues.
Response: We have done a thorough check on the overall manuscript and have amended to the best of our knowledge, the grammatical and word spelling errors, and chart formatting alignment issues. All amendments of this nature are highlighted in yellow.
- In the line 129-132 of page 3, the authors mention that “Interestingly, a coculture experiment on Adipose-derived MSC with MCF7-luminal and MDA-MB-231-basal breast cancer cells demonstrated the change towards a more dormant-epithelial phenotype associated with lower metastasis potential but higher chemoresistance [26].” I would like to ask if the authors could explain in more detail why the shift to a more dormant-epithelial phenotype would cause the cells to a higher chemoresistance.
Response: Both MCF7-luminal and MDA-MB-231-basal breast cancer cells demonstrated higher chemoresistance towards the cancer drugs - doxorubicin, tamoxifen, cisplatin and 5HNQ. It was highlighted by the authors that the decrease in metabolic and cellular activities of dormant cells had provided a form of evasion from the effects of these drugs which generally target rapidly proliferating cells. Furthermore, there was an increased expression of drug resistance related proteins shown to be triggered by MSC-exosomes in which a similar trend was reported in the study by Zhang er. Al. (2015) [Ji R, Zhang B, Zhang X, Xue J, Yuan X, Yan Y, et al. Exosomes derived from human mesenchymal stem cells confer drug resistance in gastric cancer. Cell Cycle. 2015;14(15):2473–83. We have added in the additional points in the main text as well (highlighted in green).
- In the chapter 2.2.3 of page 3-4, the end of the paragraph mentions that “these studies fell short of explaining ....”. I would like to know if the first half of the paragraph is your explanation of this issue? and if so, please adjust the order of the paragraph reasonably.
Response: The paragraph mentioned that “these studies fell short of explaining ....” have been omitted and replaced with a clearer statement: “Despite the complexities of MSC interaction with cancers in the respective TME, numerous exosomal miRNA were found to be directly associated with cancer pathways (Table 1), and were shown to be promising anti-cancer delivery agents (Table 2) with potential as noncell-based cancer therapy.”
- In the line 213-214 of page 8, there mentions “the advancement of gene engineering techniques provides opportunities to further develop”. Suggest that the authors consider where and what opportunities genetic engineering would open up and what possibilities it offers for subsequent specific applications.
Response: We have added several major examples of gene engineering and culture engineering to address the crucial role of such technology in driving the potential use of exosomal miRNA with feasible tumor targeting properties and delivery of anticancer load. These additions are highlighted in green.
- In the line 307-320 of page 10 and Table S1, The authors summarize “Roles of circulating CSC-regulating miRNAs in modulating cancer cell stemness and its linkage to disease progression and prognosis” in detail. The work is very detailed and comprehensive, but it may be worth considering a brief description in the body content to summarize some warning points for the readers.
Response: Thank you for the suggestion. We have added a paragraph at the end of Section 6.1 (highlighted red) to summarize whole section 6.1 to the readers
- At the conclusion, the authors have summarized the current progress of stem cell for cancer therapy research and some difficulties that need to be overcome, and it would be nice if the authors could offer some more constructive comments after your own consideration.
Response: Thank you for the suggestion. We have replaced the final paragraph of the conclusion with a concluding comment that reflects the overall motivating principles of all the authors in contributing towards this writeup.
- The authors may consider adding relevant references to add to the richness of this work on exosomal anticancer service studies. For example: “Progress in the research of nanomaterial-based exosome bioanalysis and exosome-based nanomaterials tumor therapy. DOI: 10.1016/j.biomaterials.2021.120873”.
Response: Thank you for the suggestion. We have cited the article in our edited draft (Section 6.3, second last two sentences, highlighted red).
Reviewer 1
The authors summarized the recent work on cancer interactions with commonly studied stem cell and their potential breakthroughs. The review is therefore timely but there are still some confusing points. I have a few comments that help to improve the clarity of the work. This review needs major revision before publication.
- The manuscript needs to be checked again for grammatical and word spelling errors, and for chart formatting alignment issues.
Response: We have done a thorough check on the overall manuscript and have amended to the best of our knowledge, the grammatical and word spelling errors, and chart formatting alignment issues. All amendments of this nature are highlighted in yellow.
- In the line 129-132 of page 3, the authors mention that “Interestingly, a coculture experiment on Adipose-derived MSC with MCF7-luminal and MDA-MB-231-basal breast cancer cells demonstrated the change towards a more dormant-epithelial phenotype associated with lower metastasis potential but higher chemoresistance [26].” I would like to ask if the authors could explain in more detail why the shift to a more dormant-epithelial phenotype would cause the cells to a higher chemoresistance.
Response: Both MCF7-luminal and MDA-MB-231-basal breast cancer cells demonstrated higher chemoresistance towards the cancer drugs - doxorubicin, tamoxifen, cisplatin and 5HNQ. It was highlighted by the authors that the decrease in metabolic and cellular activities of dormant cells had provided a form of evasion from the effects of these drugs which generally target rapidly proliferating cells. Furthermore, there was an increased expression of drug resistance related proteins shown to be triggered by MSC-exosomes in which a similar trend was reported in the study by Zhang er. Al. (2015) [Ji R, Zhang B, Zhang X, Xue J, Yuan X, Yan Y, et al. Exosomes derived from human mesenchymal stem cells confer drug resistance in gastric cancer. Cell Cycle. 2015;14(15):2473–83. We have added in the additional points in the main text as well (highlighted in green).
- In the chapter 2.2.3 of page 3-4, the end of the paragraph mentions that “these studies fell short of explaining ....”. I would like to know if the first half of the paragraph is your explanation of this issue? and if so, please adjust the order of the paragraph reasonably.
Response: The paragraph mentioned that “these studies fell short of explaining ....” have been omitted and replaced with a clearer statement: “Despite the complexities of MSC interaction with cancers in the respective TME, numerous exosomal miRNA were found to be directly associated with cancer pathways (Table 1), and were shown to be promising anti-cancer delivery agents (Table 2) with potential as noncell-based cancer therapy.”
- In the line 213-214 of page 8, there mentions “the advancement of gene engineering techniques provides opportunities to further develop”. Suggest that the authors consider where and what opportunities genetic engineering would open up and what possibilities it offers for subsequent specific applications.
Response: We have added several major examples of gene engineering and culture engineering to address the crucial role of such technology in driving the potential use of exosomal miRNA with feasible tumor targeting properties and delivery of anticancer load. These additions are highlighted in green.
- In the line 307-320 of page 10 and Table S1, The authors summarize “Roles of circulating CSC-regulating miRNAs in modulating cancer cell stemness and its linkage to disease progression and prognosis” in detail. The work is very detailed and comprehensive, but it may be worth considering a brief description in the body content to summarize some warning points for the readers.
Response: Thank you for the suggestion. We have added a paragraph at the end of Section 6.1 (highlighted red) to summarize whole section 6.1 to the readers
- At the conclusion, the authors have summarized the current progress of stem cell for cancer therapy research and some difficulties that need to be overcome, and it would be nice if the authors could offer some more constructive comments after your own consideration.
Response: Thank you for the suggestion. We have replaced the final paragraph of the conclusion with a concluding comment that reflects the overall motivating principles of all the authors in contributing towards this writeup.
- The authors may consider adding relevant references to add to the richness of this work on exosomal anticancer service studies. For example: “Progress in the research of nanomaterial-based exosome bioanalysis and exosome-based nanomaterials tumor therapy. DOI: 10.1016/j.biomaterials.2021.120873”.
Response: Thank you for the suggestion. We have cited the article in our edited draft (Section 6.3, second last two sentences, highlighted red).

Reviewer 2 Report
Major issues:
1. In conclusion part, you indicate numerous findings that have used exosomal miRNA as an alternative to cell-based cancer therapy but you do not have any references to support it.
2. Different types of cancer have different tumor microenvironment and cellular profiles. It will be better if you can discuss interactions between CSCs and exosomal miRNAs individually based on cancer types.
3. You included various types of stem cells in your review. It will be better if you just focus on one of them.
Author Response
Reviewer 2
- In conclusion part, you indicate numerous findings that have used exosomal miRNA as an alternative to cell-based cancer therapy but you do not have any references to support it.
Response: Thank you for the comment. In the conclusion part, the statement generally mentioned about the prospects of developing exosomal miRNA as an alternative to cell-based cancer therapy – “prospect of using targeted approaches such as exosomal miRNA as an alternative to cell-based cancer therapy to generate more specific positive outcomes…”. We are of the opinion that the references that were mentioned in the text (Ref 23 - Lan, T.; Luo, M.; Wei, X. Mesenchymal Stem/Stromal Cells in Cancer Therapy. J Hematol Oncol 2021, 14, 195-211) as well as the references in Table 2 are appropriate and relevant examples of promising developments of exosomal miRNA as potential noncell based cancer therapy. There was also a specific mentioned of an ongoing clinical trial in which the clinical trail reference number has been added into the text (NCT03608631) – Line 191 to 194.
- Different types of cancer have different tumor microenvironment and cellular profiles. It will be better if you can discuss interactions between CSCs and exosomal miRNAs individually based on cancer types.
Response: Thank you for the suggestion. We have added some further explanations (highlighted red) in section 6.1 to discuss the interactions between CSC and exosomal miRNAs in different cancer types/TMEs.
- You included various types of stem cells in your review. It will be better if you just focus on one of them.
Response: The focus of this review paper is to provide an overview of how studies involving major types of stem cells have addressed their respective applications as potential candidates in cancer therapy. The common theme among these different major studied types of stem cells revolves around their influence in cancer research through cell-cell interaction namely exosomal miRNA and the cell signaling mechanisms that trigger a suppression or promotion of cancer initiation or progression. By leveraging on existing findings and with current technological advancement, the application trend of cancer stem types was further addressed starting with MSC followed by iPSC and then iMSC as well as CSC. Therefore, we are of the opinion that by having all these commonly studied stem cells under a particular theme, we hope to provide a more inclusive context of how the properties of these stem cells types showed substantial prospect in their contribution toward a more targeted and personalise approach in the development of effective and feasible cancer therapy.

Round 2
Reviewer 2 Report
I think present manuscript looks better and it can be accepted.